# A substrate-trapping strategy to find E3 ubiquitin ligase substrates identifies Parkin and TRIM28 targets

Masashi Watanabe [1✉], Yasushi Saeki [2], Hidehisa Takahashi[1], Fumiaki Ohtake[3], Yukiko Yoshida[4], Yusuke Kasuga[1], Takeshi Kondo[1], Hiroaki Yaguchi[1], Masanobu Suzuki[1], Hiroki Ishida[1], Keiji Tanaka[2] & Shigetsugu Hatakeyama [1✉]

The identification of true substrates of an E3 ligase is biologically important but biochemically difficult. In recent years, several techniques for identifying substrates have been developed, but these approaches cannot exclude indirect ubiquitination or have other limitations. Here we develop an E3 ligase substrate-trapping strategy by fusing a tandem ubiquitin-binding entity (TUBE) with an anti-ubiquitin remnant antibody to effectively identify ubiquitinated substrates. We apply this method to one of the RBR-type ligases, Parkin, and to one of the RING-type ligases, TRIM28, and identify previously unknown substrates for TRIM28 including cyclin A2 and TFIIB. Furthermore, we find that TRIM28 promotes cyclin A2 ubiquitination and degradation at the G1/S phase and suppresses premature entry into S phase. Taken together, the results indicate that this method is a powerful tool for comprehensively identifying substrates of E3 ligases.

[1] Department of Biochemistry, Faculty of Medicine and Graduate School of Medicine, Hokkaido University, Kita 15, Nishi 7, Kita-Ku, Sapporo, Hokkaido 060-8638, Japan. [2] Laboratory of Protein Metabolism, Tokyo Metropolitan Institute of Medical Science, Setagaya-Ku, Tokyo 156-8506, Japan. [3] Life Science Tokyo Advanced Research Center, Hoshi University, 2-4-41 Ebara, Shinagawa-Ku, Tokyo 142-8501, Japan. [4] Ubiquitin Project, Tokyo Metropolitan Institute of Medical Science, 2-1-6, Kamikitazawa, Setagaya-Ku, Tokyo 156-8506, Japan. ✉email: mawata@med.hokudai.ac.jp; hatas@med.hokudai.ac.jp

Reversible modification of proteins by ubiquitin is one of the most important posttranslational modifications to support diverse life phenomena. Ubiquitination is catalyzed by three enzymes, ubiquitin-activating enzyme (E1), ubiquitin-conjugating enzyme (E2), and ubiquitin ligase (E3), and E3 selectively recognizes substrates[1–4]. Therefore, identifying the specific substrates of each E3 ligase and determining their ubiquitination sites are important for understanding various biological events. At least 600 E3 ligase genes exist in the human genome, and many E3 ligase genes remain to be analyzed[5,6]. There are various reasons for the technical difficulty in identifying substrates: since the interaction between a substrate and E3 is generally weak[7], there is a problem with a method based on their binding; there are several E3 ligases that ubiquitinate a substrate and conversely there are several substrates ubiquitinated by an E3 ligase[8]. Therefore, even if the E3 protein level is reduced in some way such as by gene knockdown, the ubiquitination level of the target protein may not change dramatically. There are also the problems that the amount of substrates in cells is often small and that substrates to which ubiquitin moieties serving as degradation signals are conjugated are rapidly degraded by the proteasome system[9].

Two sophisticated methods, the ligase-trapping method and the trypsin-resistant tandem ubiquitin-binding entity (TR-TUBE) method, have recently been reported[10–12]. The ligase-trapping method is a method using a probe that contains a single ubiquitin-binding domain (UBA domain) fused with a FLAG-tagged E3 ligase. This fusion method can compensate for the weakness of the binding between E3 and its substrate and can identify a specific substrate even if there is redundancy in E3 ligases that ubiquitinate a certain substrate. This probe makes it possible to identify the direct substrate of E3 by capturing the substrate, which is usually dissociated from E3 immediately after ubiquitination, via binding between the UBA domain in the probe and ubiquitin on the substrate. On the other hand, the TR-TUBE method is a method for purifying ubiquitinated peptides using a combination of TUBEs and ubiquitin remnant (K-ε-GG) antibodies[13,14]. Polyubiquitinated substrates can be protected from degradation or deubiquitination by TUBE, and it can therefore compensate for low substrate levels in cells. TUBE is composed of a certain UBA such as the human RAD23A or UBQLN1. The specificity of TUBE for the chain type of polyubiquitin is largely dependent on the specificity of a monomer. By employing a domain possessing a nonselective-binding feature, this method can identify not only substrates with ubiquitin chains involved in degradation including K11- and K48-polyubiquitin chains but also substrates with various other polyubiquitin chains including M1, K6, K27, K29, K33, and K63 chains. Furthermore, by using a ubiquitin remnant antibody, it has become possible to efficiently purify only ubiquitinated peptides.

In the ligase-trapping method, substrate trapping depends on the binding between the UBA domain and ubiquitin. Although this binding tends to be stronger than that between E3 and a substrate, it is still necessary to prepare a sample from a large amount of cells. In addition, the ligase-trapping method has no protective effect against degradation or deubiquitination of substrates, and a large amount of peptides other than ubiquitinated peptides is contaminated during MS analysis. Furthermore, in the TR-TUBE method, substrate detection becomes difficult when activity of the introduced E3 is not strong enough to overcome the amount of endogenous ubiquitinated proteins already present in the cell because the method identifies ubiquitinated proteins that increase in cells as substrate candidates by overexpression of E3. In addition, since it includes the influence of other E3 and deubiquitinating enzymes that vary depending on the introduced E3, the possibility of indirect ubiquitination cannot be excluded.

Therefore, to overcome their weaknesses, we considered the possibility of combining the advantages of the two methods.

Parkin is a causative gene for autosomal recessive juvenile Parkinsonism and is an E3 ubiquitin ligase belonging to the family of RBR (RING–IBR–RING) type E3 ligases[15–17]. Parkin is normally localized in the cytoplasm and its E3 activity is maintained in an inactive state, and it is recruited to the mitochondrial outer membrane (MOM) and activated there in response to mitochondrial depolarization[18,19]. Activated Parkin ubiquitinates primarily MOM proteins and induces mitophagy[20,21]. Tripartite motif-containing 28 (TRIM28)/KRAB-associated protein-1 (KAP1)/transcriptional intermediary factor 1β (TIF1β) is an E3 ligase belonging to the TRIM family, which is a RING-type E3 ligase family, and it has various functions such as transcription repression, transcription elongation, heterochromatin spreading, and double strand break repair in the heterochromatin region[22–27]. It has been reported that TRIM28 interacts with Krüppel-associated box zinc finger (KRAB–ZNF) transcription factors and controls the ubiquitination or stability of some of these proteins[28,29]. It is also known that the ligase activity of TRIM28 is increased by binding of certain melanoma antigen (MAGE) family members[30].

In this study, we developed a method by combining the ligase-trapping method and the TR-TUBE method and we applied the method to Parkin and TRIM28 as E3 ligases. We succeeded in identifying not only substrates reported in the past but also previously unknown substrates.

## Results

**Increase in identification of substrate candidates by fusion of TUBE with E3.** To develop a more efficient substrate identification method by combining both the advantages of effective protection of the substrate from degradation and enrichment of ubiquitinated proteins by the TR-TUBE method and direct purification of the real substrate of a ligase by the ligase-trapping method, we established a substrate identification method using a probe that fused TUBE to an E3 ligase (Fig. 1a). After introducing the probe into cells, the cells were lysed and immunoprecipitated with an anti-FLAG antibody. The protein complex captured by the probe was digested into peptides with trypsin and purified with a ubiquitin remnant antibody, and ubiquitinated peptides were identified by liquid chromatography coupled with tandem mass spectrometry (LC–MS/MS). The ubiquitinated peptides identified by the FLAG–TUBE probe not fused with an E3 ligase or the FLAG–TUBE-fused Parkin probe, which is considered to be almost inactivated under unstimulated conditions (Supplementary Data 1), and ubiquitinated peptides identified with probes fused with E3 ligase with deletion of enzyme activity were compared as negative controls to specifically determine ubiquitinated peptides identified for each E3 ligase. We compared samples with negative controls by label-free quantification (LFQ) abundance and by the total numbers of identified sequences (PSMs) of ubiquitinated peptides. Proteins with PSMs of >3 in at least one experiment (Exp) and >1 in at least two experiments were considered as substrate candidates for each E3 ligase. We first made a probe that fused TUBE to an E3 ligase (Fig. 1b). According to the method reported by Rodriguez et al., four UBA domains of the human RAD23A gene, which can bind various polyubiquitin chains, were tandemly connected, and a FLAG tag was fused to the N-terminus and an E3 ligase was fused to the C-terminus[13]. The UBA domains were connected by a flexible polyglycine linker. In order to establish a procedure for efficiently expressing the probe that we had made, we next attempted to identify substrates by transiently or stably expressing the probe in HEK293T cells. We found that substrate candidates can be

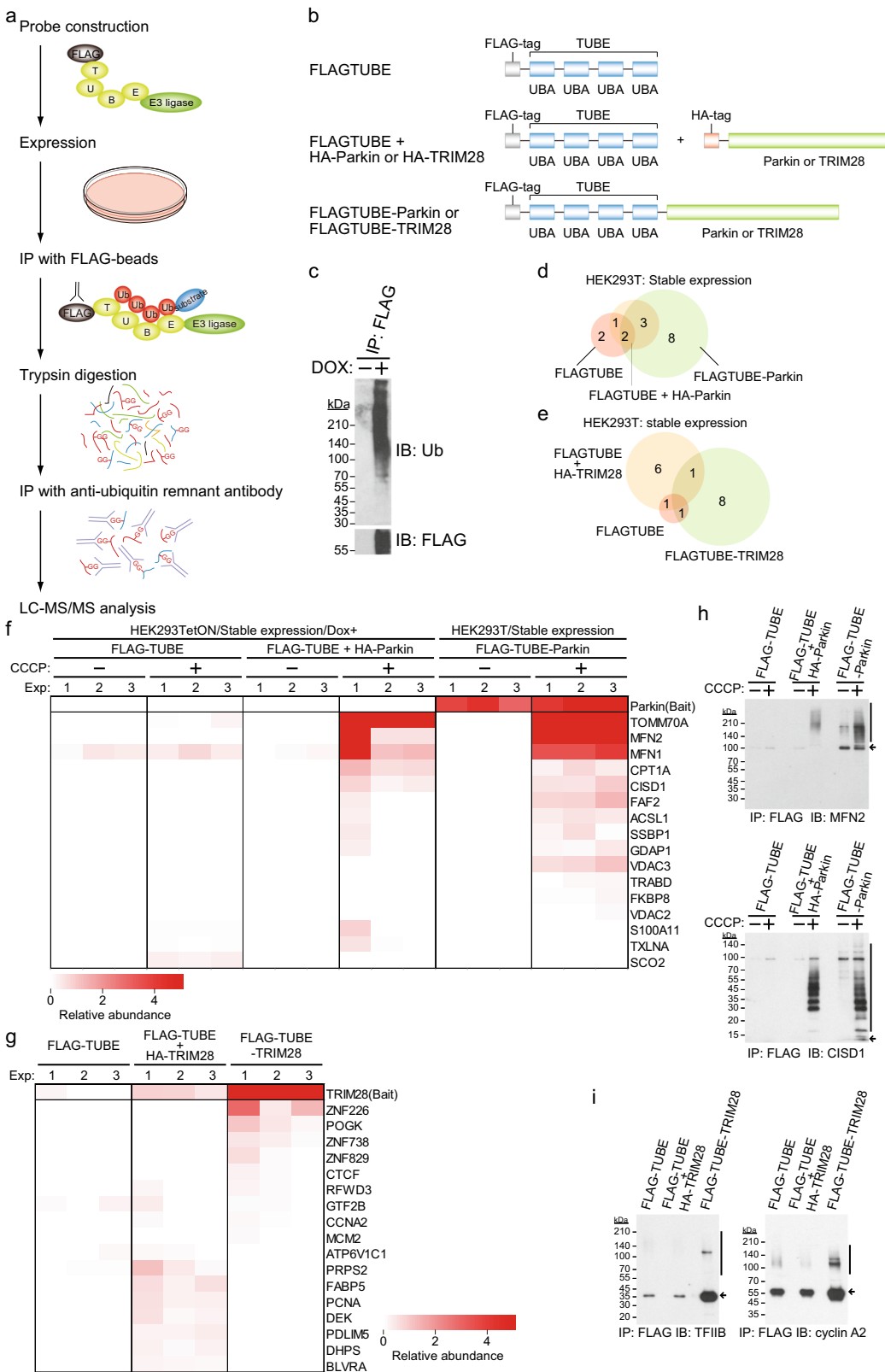

efficiently identified by stably introducing probes, and we therefore decided to use stable expression of probes in subsequent analyses (Supplementary Fig. 1).

For comparison with the TR-TUBE method, we examined whether the efficiency of identification was higher by introducing a fusion probe than by introducing TUBE and an E3 ligase independently. To verify this idea, we initially attempted to

establish a cell line stably expressing FLAG–TUBE alone, but we failed to do so. Extended expression of TUBE is known to gradually lead to cell death, and it is speculated that TUBE exerts toxicity by capturing and stabilizing ubiquitinated proteins[12]. Therefore, we next established a cell line in which the expression of FLAG–TUBE was induced in the presence of doxycycline (Fig. 1c). FLAG–TUBE was introduced into HEK293 Tet-On 3G

**Fig. 1 Fusion of TUBE with E3 increases the efficiency of identification of substrate candidates. a** Work flow for identifying substrates of an E3 ubiquitin ligase. **b** Construction of an N-terminal FLAG–TUBE-fused probe. The TUBE consists of four UBA domains of the human *RAD23A* gene with a flexible linker. **c** Establishment of HEK293 Tet-On 3G cells that inducibly express FLAG–TUBE. Cells were not treated or were treated with doxycycline (1 μg/ml) for 72 h to express FLAG–TUBE. Cell lysates were immunoprecipitated with anti-FLAG M2 agarose. The precipitates were analyzed by immunoblotting with the indicated antibodies. **d, e** Venn diagrams for identified substrate candidates in Parkin experiments with CCCP (10 μM) treatment for 1 h (**d**) and TRIM28 experiments (**e**) in HEK293T cells. To express FLAG–TUBE and each E3 ligase independently, HEK293 Tet-On 3G cells stably expressing each E3 ligase were treated with doxycycline (1 μg/ml) for 72 h to express FLAG–TUBE. HEK293 Tet-On 3G cells that inducibly express FLAG–TUBE alone were used as a control. **f, g** Substrate candidates for Parkin (**f**) and TRIM28 (**g**). Relative label-free quantification (LFQ) abundance is indicated by the color scale. Proteins with PSMs of >3 in at least one experiment (Exp) and >1 in at least two experiments were considered as substrate candidates for each E3 ligase. **h, i** Detection of ubiquitinated endogenous substrates. Cells expressing only FLAG–TUBE, FLAG–TUBE and each E3 ligase independently or the FLAG–TUBE-fused probe were harvested and anti-FLAG immunoprecipitates were analyzed by immunoblotting. Vertical bars and arrows denote the positions of ubiquitinated substrates and unmodified substrates, respectively.

cells by a retrovirus incorporating the FLAG–TUBE sequence downstream of the tetracycline responsive element, and the established cells were cultured with doxycycline for 3 days. The cells were lysed, immunoprecipitated with an anti-FLAG antibody, and analyzed by immunoblotting. We confirmed that the FLAG–TUBE probe was expressed in a doxycycline-dependent manner and that the ubiquitinated protein was coprecipitated in the presence of the FLAG–TUBE probe. Next, each E3 ligase gene was stably expressed in this cell line and analyzed by our method. Six substrate candidates were identified by the independent introduction of FLAG–TUBE and HA–Parkin, five of which were the same as those identified by the FLAG–TUBE-Parkin probe, but eight molecules that were identified by using the FLAG–TUBE–Parkin probe, including known substrates such as VDACs, were not identified by the independent introduction of FLAG–TUBE and HA–Parkin, suggesting that the substrates are more efficiently identified by the fusion probe (Fig. 1d, f). In cells in which TUBE and Parkin were independently introduced, self-ubiquitination of Parkin was not detected. Interestingly, five substrate candidates have been identified in cells in which exogenous Parkin was not introduced, and they might be molecules that undergo ubiquitination by mitochondrial depolarization, independently of Parkin. Furthermore, we analyzed TRIM28 in the same manner and identified eight substrate candidates by independent introduction of FLAG–TUBE and TRIM28. Only one substrate candidate overlapped with candidates identified by the fusion probe, and known substrates such as KRAB–ZNF proteins were not included. The fusion probe therefore appeared to be very effective (Fig. 1e, g). As well as Parkin, in cells with independent introduction, self-ubiquitination of TRIM28 was hardly detected. Consistent with the results of MS analyses, ubiquitinated substrates were moderately detected in cells with independent introduction and were highly detected in cells with fusion probes by immunoblot analyses (Fig. 1h, i). These findings indicated that the identification efficiency was dramatically increased by introducing a FLAG–TUBE-fused E3 ligase probe rather than introducing them independently.

**Increase in identification of substrate candidates by tandem connection of UBA domains**. For comparison with the ligase-trapping method, we next examined whether the TUBE-fused E3 probe can identify substrate candidates more efficiently than a single UBA domain-fused E3 probe does (Fig. 2a). We stably introduced the FLAG–UBA–Parkin probe into HEK293T cells. Eleven substrate candidates were identified, seven of which were the same as those identified by the TUBE-fused probe, but six molecules that were identified by using the TUBE-fused probe were not identified, suggesting that the substrates are more efficiently identified by using TUBE (Fig. 2b, d). Furthermore, we analyzed TRIM28 in the same manner, and four substrate

candidates were identified by the FLAG–UBA–TRIM28 probe. Only one substrate candidate overlapped with candidates identified by the TUBE-fused probe. The use of TUBE therefore appeared to be very effective (Fig. 2c, e). Consistent with the results of MS analyses, ubiquitinated substrates were more effectively detected with the use of TUBE than with the use of a single UBA domain by immunoblot analyses (Fig. 2f, g). These findings indicated that tandem connection of UBA domains dramatically increased the efficiency of substrate identification.

**Overlapping of substrate candidates between HEK293T and HeLaS3 cells**. The FLAG–TUBE–E3 ligase probe was also stably introduced into HeLaS3 cells. Ser 65-phosphorylation of Parkin during CCCP treatment was confirmed by MS analysis (Supplementary Fig. 2b). Thirty substrate candidates were identified by using the FLAG–TUBE–Parkin probe in HeLaS3 cells (Supplementary Fig. 2a, c, and Supplementary Data 2). Of those, 13 molecules were identical to those identified in HEK293T cells. We could identify more substrate candidates of Parkin in HeLaS3 cells than in HEK293T cells. On the other hand, one substrate candidate was identified using the FLAG–TUBE-TRIM28 probe in HeLaS3 cells. Therefore, more substrate candidates of TRIM28 could be identified in HEK293T cells than in HeLaS3 cells (Supplementary Fig. 2d, e). The results of MS analyses were also consistent with the results of immunoblot analysis (Supplementary Fig. 2f, g). Taken together, the results indicated that different substrate candidates are identified depending on the cell context. These results may reflect the differences in substrates that are expressed in each cell and the difference in the activation mechanisms of E3 ligases. Intriguingly, it has been reported that the E3 ligase activity of TRIM28 is greatly increased in the presence of some MAGE proteins and that there is a difference in the type of MAGE proteins expressed in each cell[30]. Our results may indicate that MAGE proteins expressed in HEK293T cells induce greater activation of TRIM28 than do those expressed in HeLaS3 cells.

**Difference in substrate candidates by difference in fusion sites between TUBE and E3**. To explore the difference in substrates due to the fusion site of the probes, we next compared the FLAG–TUBE–Parkin probe in which TUBE was fused to the N-terminus of Parkin with the Parkin–FLAG–TUBE probe in which TUBE was fused to the C-terminus of Parkin (Fig. 3a). Six substrate candidates were identified using the Parkin–FLAG–TUBE probe in HeLaS3 cells, and all of them were included in the substrate candidates identified by the FLAG–TUBE–Parkin probe (Fig. 3b, d, f). It is known that there are E3 ligases for which activities are impaired by C-terminal extensions[31]. The catalytic domain of HECT-type E3 ligases such as human E6AP and yeast Rsp5p is located at the C-terminus, and its enzymatic activity is

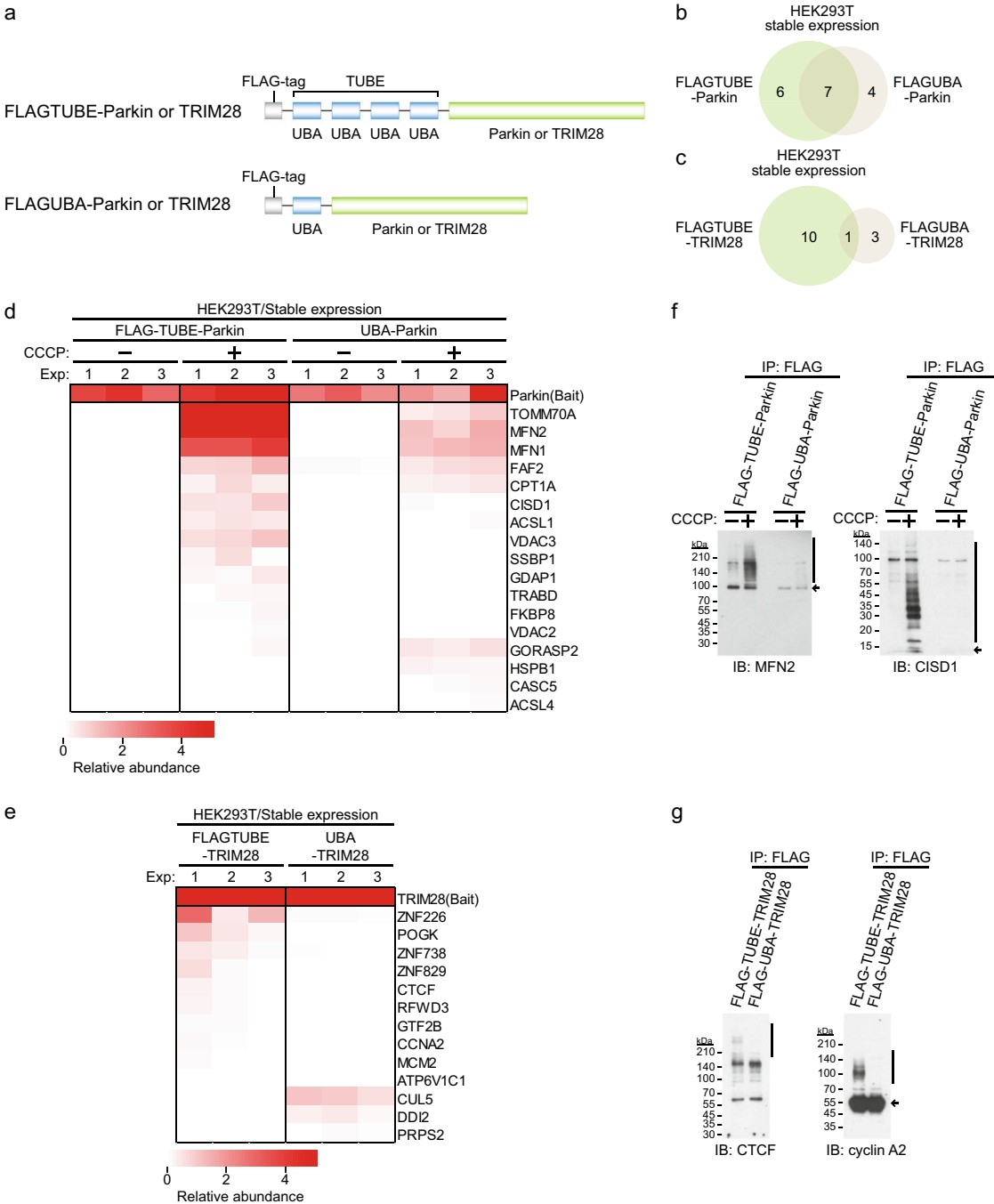

**Fig. 2 Tandem connection of UBA domains increases the efficiency of identification of substrate candidates. a** Construction of an N-terminal FLAG–UBA-fused probe. **b**, **c** Venn diagrams for identified substrate candidates in Parkin experiments with CCCP (10 μM) treatment for 1 h (**b**) and TRIM28 experiments (**c**) in HEK293T cells stably expressing each probe. Probes in which TUBE or UBA was fused to the N terminus of each E3 ligase were used. **d**, **e** Substrate candidates for Parkin (**d**) and TRIM28 (**e**). Relative label-free quantification (LFQ) abundance is indicated by the color scale. Proteins with PSMs of >3 in at least one experiment (Exp) and >1 in at least two experiments were considered as substrate candidates for each E3 ligase. **f**, **g** Detection of ubiquitinated endogenous substrates. Cells expressing the FLAG–TUBE or FLAG–UBA-fused probe were harvested and anti-FLAG immunoprecipitates were analyzed by immunoblotting. Vertical bars and arrows denote the positions of ubiquitinated substrates and unmodified substrates, respectively.

interfered with when the C terminus is extended. Since the catalytic domain of Parkin is also located at the C-terminus, C-terminal FLAG–TUBE fusion may also interfere with the enzymatic activity. Likewise, we next compared FLAG–TUBE–TRIM28 with TRIM28–FLAG–TUBE (Fig. 3a, c, e, g). We could identify more substrate candidates by using the C-terminal TUBE-tagged TRIM28

than by using the N-terminal TUBE-tagged TRIM28. These findings suggest that it is necessary to fuse TUBE in a way that does not affect the activity of each E3 ligase.

**Validation and landscape of the substrate candidates for Parkin and TRIM28 E3 ligases.** In order to validate the obtained

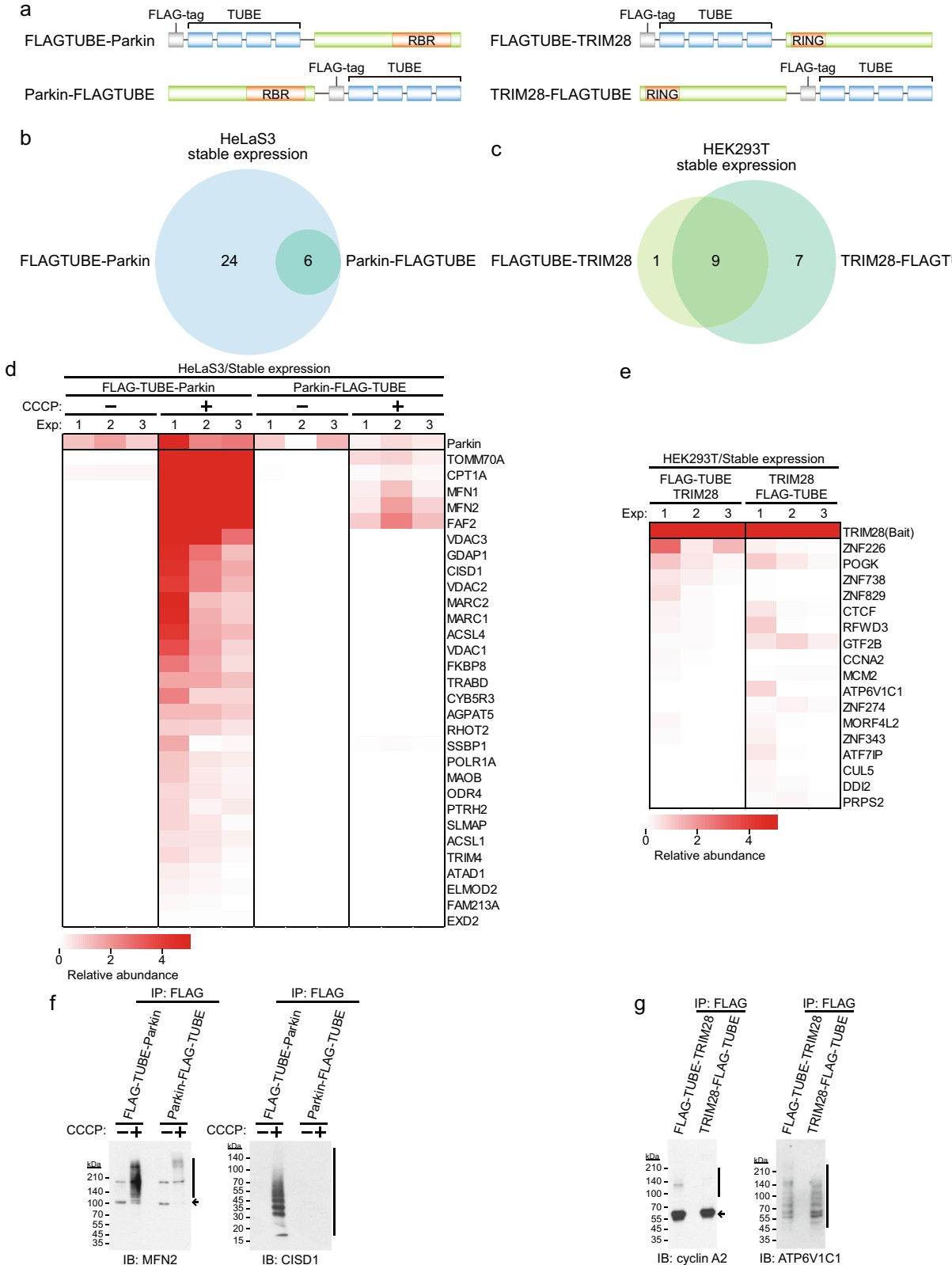

substrate candidates, intracellular ubiquitinated proteins were pulled down with a recombinant TUBE protein prepared in an *Escherichia coli* expression system and evaluated by immunoblot analysis with an antibody for each substrate (Fig. 4a, b). HA–Parkin was stably expressed in HEK293T cells, and mitochondrial depolarization was induced by CCCP. Ubiquitinated proteins were pulled down with a recombinant

GST–FLAG–TUBE protein and the precipitates were analyzed by immunoblotting. We then confirmed that ubiquitination of the candidates, CISD1, CPT1A, and MFN2, could be detected only in cells stably expressing HA–Parkin and treated with CCCP (Fig. 4c). Furthermore, TRIM28 was stably knocked down in HEK293T cells, and ubiquitinated proteins were pulled down with recombinant FLAG–TUBE protein, and then the

**Fig. 3 Efficiency of detection of substrate candidates by difference in the fusion site between TUBE and E3. a** Construction of an N-terminal or C-terminal FLAG–TUBE-fused probe. **b, c** Venn diagram for identified substrate candidates in Parkin experiments with CCCP (10 μM) treatment for 1 h in HeLaS3 cells stably expressing each probe (**b**) and in TRIM28 experiments in HEK293T cells stably expressing each probe (**c**). A probe in which TUBE was fused to the N or C terminus of Parkin or TRIM28 was used. **d, e** Substrate candidates for Parkin (**d**) and TRIM28 (**e**). Relative label-free quantification (LFQ) abundance is indicated by the color scale. Proteins with PSMs of >3 in at least one experiment (Exp) and >1 in at least two experiments were considered as substrate candidates for each E3 ligase. **f, g** Detection of ubiquitinated endogenous substrates. Cells expressing the N-terminal or C-terminal FLAG–TUBE-fused probe were harvested and anti-FLAG immunoprecipitates were analyzed by immunoblotting. Vertical bars and arrows denote the positions of ubiquitinated substrates and unmodified substrates, respectively.

immunoprecipitates were analyzed by immunoblotting in the same way. TRIM28 knockdown reduced the amount of ubiquitinated candidate proteins, TFIIB, ATP6V1C1, and cyclin A2 (Fig. 4d and Supplementary Fig. 3). The results indicate that the candidates identified by the method using the E3 probe fused with TUBE are *bona fide* substrates that undergo ubiquitination by each E3 ligase. We also evaluated some candidates identified only by FLAG-monoUBA-fused E3 or only by independent introduction of TUBE and E3 (Supplementary Fig. 4). Among those we examined, the amount of ubiquitinated DDI2 was reduced by TRIM28 knockdown.

Supplementary Figure 5a, b shows summaries of the results of Parkin and TRIM28 analyses. Functional annotation analysis using the DAVID resource was performed for the substrate candidate proteins identified with the FLAG–TUBE–Parkin probe in HeLaS3 cells, for which the most candidates were obtained. The most significantly enriched GO terms were "mitochondrial outer membrane" ($p = 2.6E−20$) and "mitochondrion" ($p = 9.7E−13$) in the cellular component (GOCC) group, being consistent with the localization of activated Parkin upon mitochondrial damage (Supplementary Fig. 5c)[32,33].

The analysis was also performed for the substrate candidate proteins identified with the TRIM28–FLAG–TUBE probe in HEK293T cells. Significantly enriched terms were "regulation of transcription, DNA-templated" ($p = 4.0E−3$) in the GO biological process group and "Krüppel-associated box" ($p = 1.4E−4$) in the InterPro database (Supplementary Fig. 5d, e). TRIM28 has been identified as a transcription repressor and a KRAB–ZNF-binding protein. KRAB–ZNF proteins are transcription factors that regulate the expression of target genes, and TRIM28 mediates ubiquitin conjugation or stability of some of these proteins[28,29]. Therefore, the results of our study are consistent with previous reports.

**TRIM28 knockdown stabilizes cyclin A2 at the G1/S phase, while it stabilizes TFIIB independently of the cell cycle.** To biologically investigate the effect of ubiquitination by TRIM28 of cyclin A2 and TFIIB, which have not been reported as substrates in the past, protein levels of the substrates were examined by immunoblot analysis. The amount of cyclin A2 and TFIIB proteins was increased by TRIM28 knockdown (Fig. 5a). Next, we examined the protein stability of cyclin A2 and TFIIB. TRIM28-knockdown cells and corresponding control cells were treated with cycloheximide for the indicated times. TRIM28 knockdown considerably suppressed the degradation of TFIIB protein and slightly suppressed the degradation of cyclin A2 (Fig. 5b). Since it is known that the amount of cyclin A2 is regulated depending on the cell cycle, the same experiments were performed at the G1/S and G2/M phases. When cells were treated with aphidicolin for 36 h and then treated with cycloheximide for the indicated times, TRIM28 knockdown also suppressed the degradation of TFIIB and cyclin A2 proteins (Fig. 5c). On the other hand, when cells were treated with nocodazole for 14 h and then treated with cycloheximide for the indicated times, TRIM28 knockdown inhibited the degradation of TFIIB protein but did not affect the

degradation of cyclin A2 protein (Fig. 5d). To understand this cell cycle-dependent regulation of cyclin A2 protein by TRIM28, we examined the binding of TRIM28 to cyclin A2 at each cell cycle stage. We found that the amount of cyclin A2 bound to TRIM28 increased stoichiometrically in accordance with the amount of cyclin A2 expression (Fig. 5e). Meanwhile, aphidicolin treatment decreased the stability of TRIM28, whereas asynchronous or nocodazole treatment did not affect it, suggesting that the expression level of TRIM28 is dependent on the phases of the cell cycle (Fig. 5b–d). Taken together, the results indicate that TRIM28 leads TFIIB to ubiquitin-mediated degradation regardless of the stage of the cell cycle, whereas TRIM28 leads cyclin A2 to degradation mainly at the G1/S phase. TRIM28-mediated regulation of cyclin A2 appears to be dependent on the amount of cyclin A2 protein bound to TRIM28 at each cell cycle stage.

**TRIM28 is required to repress cyclin A and prevent premature entry into the S phase.** It has been reported that accumulated cyclin A has S-phase-promoting activity[34,35]. Knockdown of FZR1 and UBE2C, which are an E3 ubiquitin ligase and an E2 conjugating enzyme for cyclin A, respectively, also shortens the G1 phase by increasing cyclin A level[36–38]. We therefore examined whether a similar phenotype is observed by TRIM28 depletion. U2OS cells were treated for 14 h with nocodazole, harvested by mitotic shake-off, released from a mitotic block, and analyzed by flow cytometry after DNA staining. We found that TRIM28 knockdown slightly promoted S phase entry (Fig. 6a, b). This result is consistent with the accumulation of cyclin A2, suggesting that TRIM28 is involved in the regulation of S phase progression via cyclin A2.

**Discussion**
In this study, we combined two pioneering methods to take advantages of both methods and to overcome their weaknesses, and we established a method that can more efficiently identify substrates of E3 ligases. When we performed analysis with Parkin, we identified 37 candidates and 131 ubiquitination sites. The number of substrates identified in this study was larger than that in a previous study (Figs. 1d, 2b, and Supplementary Data 2). Moreover, we performed analysis with TRIM28, and we identified 26 substrates and 103 ubiquitination sites that were not identified by past methods (Figs. 1e, 2c, and Supplementary Data 3).

Compared with fused probes, when FLAG–TUBE and E3 ligase were independently introduced into cells, not only did the number of substrate candidates decrease but also the ubiquitination of E3 ligase itself could hardly be detected (Fig. 1f, g). In the case of Parkin, five substrate candidates identified by independent introduction were the same as those identified by the fusion probe, but eight molecules identified by using the fusion probe were not identified by independent introduction. In the case of TRIM28, only one candidate identified by independent introduction was the same as that identified by the fusion probe, but nine molecules identified by using the fusion probe were not identified by independent introduction. It is known that independently introduced FLAG–TUBE captures intracellular

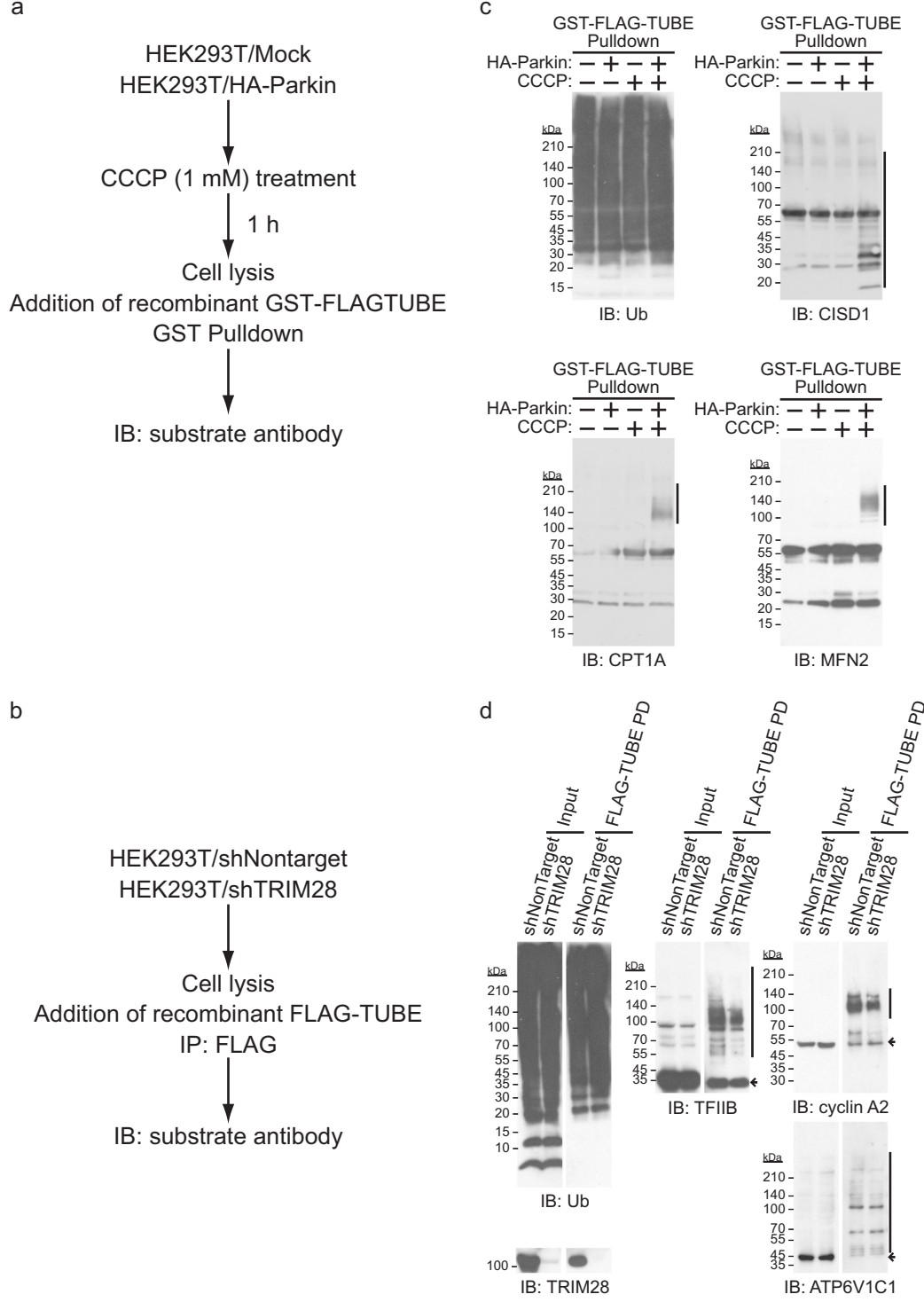

**Fig. 4 Validation of the substrate candidates for Parkin and TRIM28 E3 ligases. a**, **b** Schematic presentations indicating the steps in the ubiquitination assay using recombinant GST–FLAG–TUBE (**a**) or FLAG–TUBE proteins (**b**). **c** HEK293T cells stably expressing HA–Parkin or harboring an empty vector (Mock) were not treated with or were treated with CCCP (10 μM) for 1 h. Cell lysates were pulled down with the beads prebound by GST–FLAG–TUBE proteins. The precipitates were analyzed by immunoblotting. Vertical bars and arrows denote the positions of ubiquitinated substrates and unmodified substrates, respectively. **d** HEK293T cells with stable knockdown of TRIM28 or corresponding control cells were harvested. Cell lysates were pulled down with the beads prebound by FLAG–TUBE proteins. The precipitates were analyzed by immunoblotting. Vertical bars and arrows denote the positions of ubiquitinated substrates and unmodified substrates, respectively.

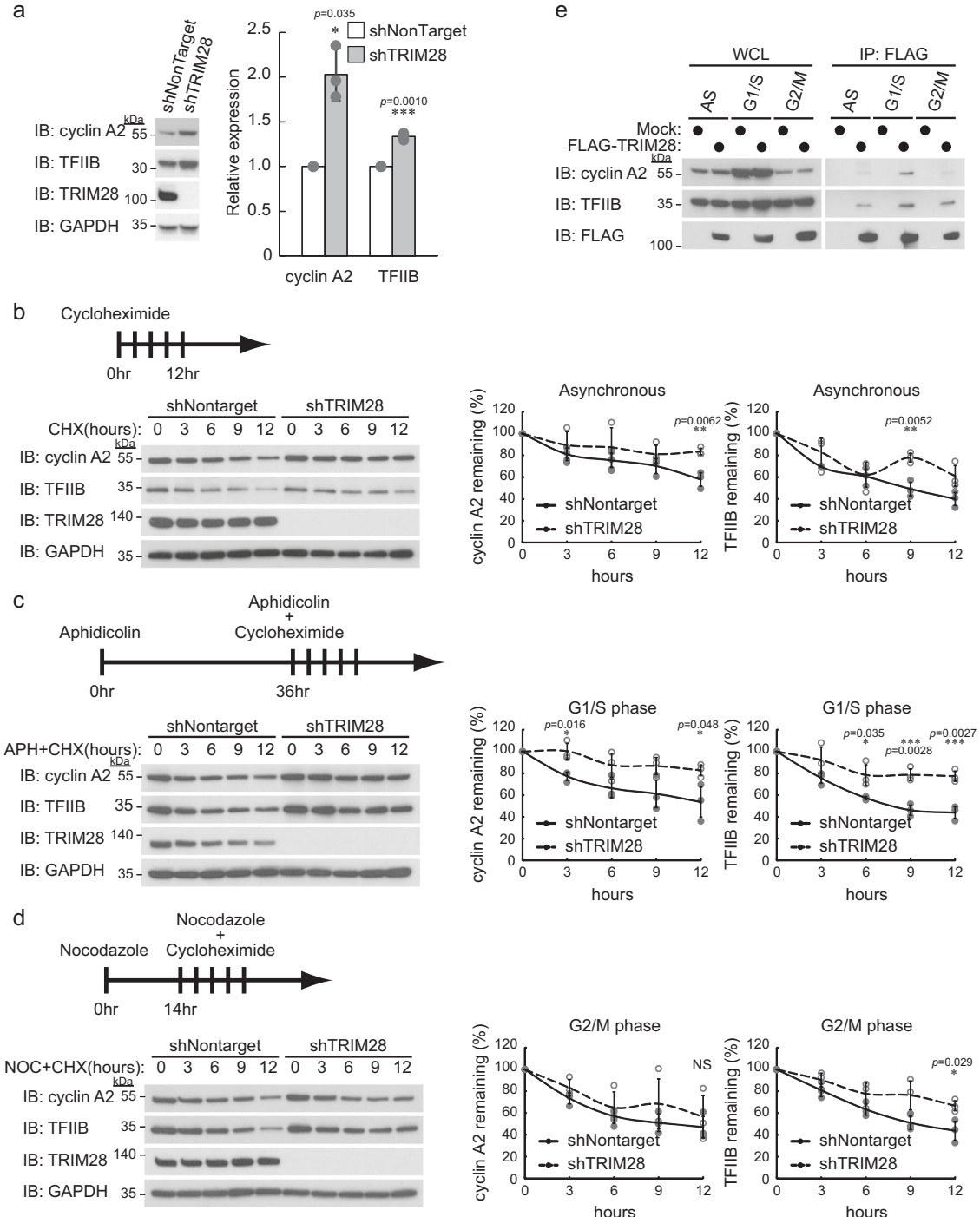

**Fig. 5 TRIM28 destabilizes TFIIB independently of the cell cycle stage and destabilizes cyclin A2 specifically at the G1/S phase. a** Immunoblot analysis of cyclin A2 and TFIIB proteins in TRIM28-knockdown HEK293T cells (left). The intensity of the cyclin A2 and TFIIB bands was normalized to that of the corresponding GAPDH bands and is indicated as a relative intensity of the normalized value of control shRNA-treated cells (right). The data represent means ± s.d. from three independent experiments. The p values for the indicated comparisons were determined using Student's t test (*p < 0.05; ***p < 0.005). **b–d** Cycloheximide-chase assay of asynchronous (**b**), G1/S phase (**c**), and G2/M phase (**d**) cells. HEK293T cells with stably knocked down TRIM28 or the corresponding control cells were cultured in the presence of cycloheximide (50 μg/ml) for the indicated times. For G1/S or G2/M synchronization, cells were treated for 36 h with aphidicolin (4 μg/ml) or for 14 h with nocodazole (400 ng/ml). The data represent means ± s.d. from three independent experiments. The p values for the indicated comparisons were determined using Student's t test (*p < 0.05; **p < 0.01; ***p < 0.005). **e** In vivo assay for interaction between TRIM28 and cyclin A2 or TFIIB. HEK293T cells stably expressing FLAG-TRIM28 were generated. Asynchronous, G1/S phase, and G2/M phase cells were harvested. Whole cell lysates were immunoprecipitated with anti-FLAG antibody, and then immunoblot analysis was performed with the indicated antibodies.

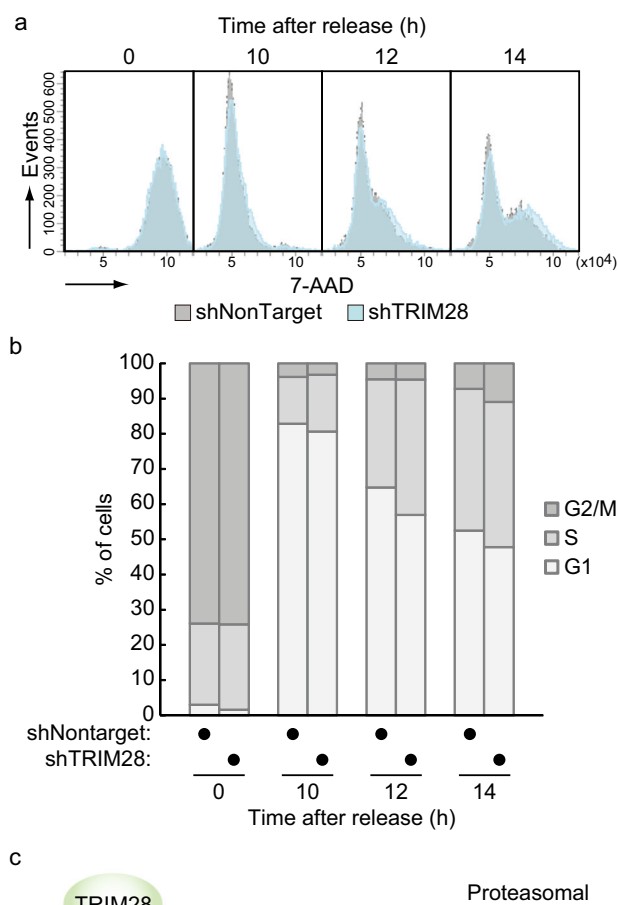

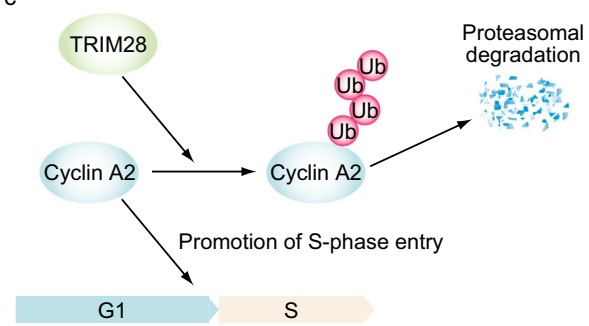

**Fig. 6 TRIM28 knockdown leads to premature entry into S phase. a**, **b** U2OS cells with stably knocked-down TRIM28 or the corresponding control cells were treated for 14 h with nocodazole (400 ng/ml) and harvested by mitotic shake-off, washed three times in a medium and replated. Cells were harvested at the indicated time points. Samples were analyzed by 7-AAD staining and flow cytometry. A histogram (**a**) and a cell cycle distribution (**b**) are shown. The results are representative of three independent experiments. **c** A schematic model of TRIM28-mediated suppression of S-phase progression via cyclin A2 protein regulation. When TRIM28 does not exist at the G1/S phase, cyclin A2 accumulates, leading to premature S-phase entry. TRIM28 eliminates excess cyclin A2 and regulates proper S-phase progression.

ubiquitinated proteins without distinction. Therefore, to identify substrates and self-ubiquitination of E3 by the method using FLAG–TUBE and an E3 ligase independently, the amount of ubiquitinated substrates should be sufficient to overcome the amount of ubiquitinated proteins already present in cells. When mitochondrial depolarization is induced by CCCP, Parkin is highly activated and promiscuously targets multiple substrates at the outer mitochondrial membrane. FLAG–TUBE independently introduced to Parkin can therefore overcome the amount of pre-

existing ubiquitinated proteins to some extent. On the other hand, the results from TRIM28 indicated that it may be difficult to identify substrates of E3 ligases with relatively low activity by independent introduction. Since the fusion probe binds mainly to a ubiquitinated protein existing in the vicinity of its E3 ligase, it is highly probable that the substrate binds to the probe immediately after the substrate is ubiquitinated by a specific E3 ligase. Therefore, even if the amount of substrates of an E3 ligase comprise a small population in the whole ubiquitinated protein pool in cells, specific substrates can be identified by this method.

In the ligase-trapping method, by fusing the ubiquitin-binding domain with an E3 ligase, the substrate should be trapped immediately after ubiquitination of the substrate by an E3 ligase. Therefore, one of the advantages of using this probe is that it can identify specific substrates even if the substrates are redundantly targeted by other E3 ligases in cells. In this study, we used TUBE, which has an approximately 1000-times higher affinity than that of one of the UBAs, the UBA domain[13]. The method also has the advantage of ubiquitinated proteins binding to TUBE in cells being stabilized. This protective effect may be due to the inhibition of binding of deubiquitinating enzymes or ubiquitin receptors involved in the proteasome pathway. Actually, tandem connection of UBA domains dramatically increased the efficiency of substrate identification, indicating that these advantages contribute to efficient concentration of specific ubiquitinated substrates of each E3 ligase (Fig. 2d, e).

Of the peptides derived from the substrate, ubiquitinated peptides become direct evidence as the substrate and are the most important information in the E3-substrate relationship. Other peptides may not be necessary for identification of the substrate as long as information on ubiquitinated peptides can be obtained upon identification. When the complex captured by FLAG–TUBE is trypsinized into peptides and then purified with a ubiquitin remnant antibody, it is possible to eliminate peptides that have not undergone ubiquitination in the substrate and identify only peptides that have undergone ubiquitination. As a result, more efficient MS analysis can be performed, and the ubiquitination site of the substrate can also be determined simultaneously.

In this study, we screened substrates with a probe in which TUBE were fused to an E3 ligase at its amino or carboxyl terminus. It has been reported that epitope tagging of some E3 ligases affects the activity and stability of E3s. In Parkin, N-terminal epitope tagging results in release of the autoinhibition by its Ubl domain and increase of its enzymatic activity[39,40]. The use of CCCP may also induce artificial activation, far from physiological conditions. Therefore, substrate candidates identified by our method should be carefully validated and interpreted, considering various factors such as modifications of bait proteins and artificial stimulations in cells.

We found that TRIM28 ubiquitinates and degrades TFIIB and cyclin A2 as substrates and that cyclin A2 was especially degraded in the G1/S phase by TRIM28. As a result, premature entry into S phase was observed in TRIM28-knockdown cells. This phenotype in cells depleted of TRIM28 that was found in this study markedly resembles the phenotype of cells lacking FZR1 or UBE2C, which are an E3 ubiquitin ligase and an E2 conjugating enzyme for cyclin A, respectively. Taken together, the results indicate that TRIM28 regulates the appropriate progression into S phase by degrading cyclin A2.

In summary, our method has various advantages for identifying a specific substrate for each E3 ligase. By using our method, it will be possible to comprehensively identify the substrate of a specific E3 ligase, even an E3 ligase with relatively low activity, and our method will help to clarify the mechanism of the function exerted by an E3 ligase.

## Methods

**Cell culture.** HEK293T, HEK293 Tet-On 3G (Takara, Shiga, Japan) and HeLaS3 cells were cultured under an atmosphere of 5% $CO_2$ at 37 °C in Dulbecco's modified Eagle's medium (Sigma-Aldrich, St. Louis, MI) supplemented with 10% fetal bovine serum (FBS) (Gibco BRL, Paisley, UK).

**Cloning of cDNAs and plasmid construction.** Tandem-repeated ubiquitin-binding entity (TUBE) containing four tandem UBA domains of the human *RAD23A* gene was cloned with a FLAG tag at its amino terminus in pQCXIP (Takara, Shiga, Japan), pRetro-TRE3G (Takara) and pGEX-6P-1 (GE Healthcare, Uppsala, Sweden). Tandem UBA domains were ligated using *Bgl*II and *Bam*HI restriction sites to link continuously with a polyglycine linker between the two domains. Human and mouse ubiquitin ligase cDNAs were cloned in pQCXIP containing a FLAG tag at its amino or carboxyl terminus and in the pMXs-IRES-Bsd vector (Cell Biolabs, Inc., San Diego, CA) containing an HA-tag at its amino terminus.

**Transfection.** HEK293T cells were transfected using FuGENE HD (Promega Corporation, Madison, WI). After 48 h, the cells were harvested.

**Retrovirus expression system.** Retrovirus vectors were transfected with the pCL10A1 vector (Novus Biologicals) into HEK293T cells to generate recombinant retroviruses. HEK293T cells or HeLaS3 cells were infected with the recombinant retroviruses and selected in a medium containing puromycin (5 μg/ml or 2 μg/ml, respectively, Thermo Fisher Scientific, Lafayette, CO) or Blasticidin S (10 μg/ml, FUJIFILM Wako Chemicals, Osaka, Japana).

**RNA interference.** GIPZ human lentiviral short hairpin RNA (shRNA) clones (non-silencing verified negative control and TRIM28 (V3LHS_640068)) were purchased from Dharmacon (Horizon Discovery, Cambridge, UK). A lentiviral vector system with a murine stem cell virus promoter was kindly provided by St. Jude Children's Research Hospital. To produce shRNA lentiviral particles, a four-plasmid mixture consisting of pCAG-kGP4.1R (6 μg), pCAG4-RTR2 (2 μg), pCAGGS-VSVG (2 μg), and pGIPZ-based lentivector plasmids (10 μg) was cotransfected into approximately 50% confluent HEK293T cells in 100-mm dishes using Fugene HD (Promega Corporation, Madison, WI). Four hour after transfection, the culture medium was replaced. The culture supernatant containing the lentivirus was collected 48 h after transfection, filtered through 0.45-μm membranes, and concentrated using an Amicon Ultra filter unit 30k (Merck, Darmstadt, DE).

**Immunoblot analysis.** Immunoblot analysis was performed with primary antibodies, horseradish peroxidase-conjugated antibodies to mouse or rabbit immunoglobulin G (1:2500 dilution in TBST containing 3% skim milk, Promega) and enhanced chemiluminescence (ECL) reagents (Pierce ECL Western Blotting Substrate and SuperSignal West Dura Extended Duration Substrate, Thermo Scientific). The following primary antibodies were used: anti-FLAG M2 (Sigma-Aldrich, 1:2000 dilution), anti-MFN2 (12186-1-AP, Proteintech, Rosemont, IL, 1:2000 dilution), anti-CISD1 (16006-1-AP, Proteintech, Rosemont, IL, 1:2000 dilution), anti-CPT1A (15184-1-AP, Proteintech, 1:2000 dilution), anti-CTCF (A300-543A, Bethyl Laboratories, Montgomery, TX, 1:2000 dilution), anti-ATP6V1C1 (16054-1-AP, Proteintech, 1:2000 dilution), anti-TFIIB (C-18, sc-225, Santa Cruz, Dallas, TX, 1:500 dilution), anti-cyclin A2 (18202-1-AP, Proteintech, 1:2000 dilution), anti-TRIM28 (ab10483, abcam, 1:10000 dilution), anti-GAPDH (AM4300, Invitrogen, 1:20000 dilution), HSP90 (610419, BD Biosciences, San Jose, CA, USA, 1:2000 dilution), anti-PRPS1/2/3 (A-11, sc-376440, SCBT, 1:500 dilution), anti-CUL5 (sc-13014, SCBT, 1:500 dilution), anti-ACSL4 (22401-1-AP, Proteintech, 1:2000 dilution), anti-DDI2 (A-3, sc-514004, SCBT, 1:500 dilution), anti-PCNA (PC10, sc-56, SCBT, 1:500 dilution), anti-DHPS (A-10, sc-365077, SCBT, 1:500 dilution), anti-BLVRA (F-1, sc-393385, SCBT, 1:500 dilution), and anti-Ub (P4D1, sc-8017, SCBT, 1:200 dilution).

**Immunoprecipitation of probe-bound proteins, trypsin digestion, and immunoprecipitation of diGly peptides.** Cells ($5.0 \times 10^7$) were lysed in a solution containing 50 mM Tris-HCl (pH 7.6), 300 mM NaCl, 10% glycerol, 0.2% NP-40, 10 mM iodoacetamide (Sigma-Aldrich), 10 mM N-ethylmaleimide (Sigma-Aldrich), 0.5 mM 4-(2-aminoethyl)-benzenesulfonyl fluoride hydrochloride (AEBSF, Roche, Branchburg, NJ), 10 μM MG132 (Merck, Darmstadt, DE), and PhosStop phosphatase inhibitors (Sigma-Aldrich). The cell lysates were sonicated and centrifuged at 16,000×*g* for 10 min at 4 °C, and the resulting supernatant was incubated with anti-FLAG M2 agarose (Sigma-Aldrich) for 2 h at 4 °C. The resin was separated by centrifugation, washed five times with ice-cold lysis buffer, and subjected to on-bead trypsin digestion (Promega). After tryptic digestion, the samples were acidified with TFA and desalted by solid-phase extraction using GL-Tip GC and GL-Tip SDB (GL Sciences, Tokyo, Japan). Eluted peptides were dried by vacuum centrifugation, dissolved in 100 μL of immunoaffinity purification (IAP) buffer, and incubated for 2 h at 4 °C with 20 μL of PTMScan Ubiquitin Branch Motif (K-ε-GG) Immunoaffinity Beads (Cell Signaling Technology). The

beads were washed twice with 500 μL of IAP buffer and three times with 500 μL of distilled water, and peptides were eluted twice with 25 μL of 0.15% TFA. The eluted peptides were desalted using GL-Tip SDB and GL-Tip GC before LC–MS analysis.

**Mass spectrometry analysis of diGly peptides.** Desalted tryptic digests were analyzed by nanoflow ultra-HPLC (EASY-nLC 1000; Thermo Fisher Scientific) on-line coupled to an Orbitrap Elite instrument (Thermo Fisher Scientific). The mobile phases were 0.1% formic acid in water (solvent A) and 0.1% formic acid in 100% acetonitrile (solvent B). Peptides were directly loaded onto a C18 Reprosil analytical column (3 μm in particle size, 75 μm in i.d., and 12 cm in length; Nikkyo Technos, Tokyo, Japan) and separated using a 150-min two-step gradient (0–35% for 130 min, 35–100% for 5 min, and 100% for 15 min of solvent B) at a constant flow rate of 300 nL/min. For ionization, a liquid junction voltage of 1.6 kV and a capillary temperature of 200 °C were used. The Orbitrap Elite instrument was operated in the data-dependent MS/MS mode using Xcalibur software (Thermo Fisher Scientific) with survey scans acquired at a resolution of 120,000 at m/z 400. The top 10 most abundant isotope patterns with a charge ranging from 2 to 4 were selected from the survey scans with an isolation window of 2.0 m/z and fragmented by collision-induced dissociation with normalized collision energies of 35. The maximum ion injection times for the survey and MS/MS scans were 60 ms, and the ion target values were set to 1e6 for the survey and MS/MS scans. Ions selected for MS/MS were dynamically excluded for 60 s for diGly peptide identification.

**Protein identification from MS data.** Proteome Discoverer software (version 2.4; Thermo Fisher Scientific) was used to generate peak lists. The MS/MS spectra were searched against a UniProt Knowledgebase (version 2015_09) using the SequestHT search engine. The precursor and fragment mass tolerances were set to 10 ppm and 0.6 Da, respectively. Methionine oxidation, protein amino-terminal acetylation, Asn/Gln deamidation, Ser/Thr/Tyr phosphorylation, diglycine modification of Lys side chains, and Cys carboxymethyl modification were set as variable modifications for database searching. Peptide identification was filtered at a 1% false-discovery rate. To identify specific substrates of ubiquitin ligases, the results of three individual samples (cells expressing FLAG–TUBE fused with WT ubiquitin ligase or dominant-negative mutant ubiquitin ligase and cells treated with CCCP (10 μM) or not treated) were assembled into one multiconsensus report using Proteome Discoverer software. We compared samples with negative controls by LFQ abundance (2.7 times more for TRIM28 and 5 times more for Parkin) and by the total numbers of identified sequences (PSMs) of ubiquitinated peptides (2.7 times more for TRIM28 and 10 times more for Parkin). Proteins with PSMs of >3 in at least one experiment (Exp) and >1 in at least two experiments were considered as substrate candidates for each E3 ligase.

**Functional enrichment analysis.** For the list of substrate candidates, we searched for statistically enriched Gene Ontology (GO) terms and InterPro domains using the DAVID database (https://david.ncifcrf.gov/), and the background was set to the entire human genome data set. Enrichment *P* values were calculated using Fisher's exact test. The threshold level for all functional enrichment analyses was set for *P* values < 0.05 and contributing rate of proteins of >25% (for Parkin) or of >15% (for TRIM28). The terms were ranked according to their enrichment *P* values.

**Recombinant protein purification and FLAG–TUBE immunoprecipitation.** Recombinant GST-fused FLAG–TUBE protein was expressed in *Escherichia coli* strain DH5α for 16 h with 0.2 mM isopropyl-1-thio-β-d-galactopyranoside. One liter of bacterial culture medium was centrifuged, and the pellet was resuspended in 40 mL of phosphate- buffered saline (PBS) (Thermo Fisher Scientific) supplemented with AEBSF (Roche, Branchburg, NJ) and disrupted with a French pressure cell (Aminco, Thermo Fisher Scientific). The lysates were clarified by ultracentrifugation, and the supernatants were incubated on ice for 2 h after the addition of 1 mL of glutathione sepharose 4B beads (GE Healthcare, Uppsala, Sweden). The beads were washed 5 times in PBS and the bound proteins were eluted by 10 mM reduced glutathione or treated with PreScission protease (GE Healthcare) at 4 °C overnight. The excised GST and PreScission protease were removed from FLAG–TUBE by repurification on glutathione-Sepharose 4B beads. For FLAG–TUBE immunoprecipitation, 5 μg of GST–FLAG–TUBE was bound to Glutathione Sepharose beads or 5 μg of FLAG–TUBE was bound to anti-FLAG M2 agarose (Sigma-Aldrich) (20 μl for each sample, respectively) for 1 h, and the beads were then washed three times and resuspended in ice-cold lysis buffer. Cell lysates were incubated with the beads prebound by GST–FLAG–TUBE or FLAG–TUBE proteins for 2 h at 4 °C on a rotating platform. Nonspecific binding was removed by washing five times with ice-cold lysis buffer. The proteins bound to GST–FLAG–TUBE and FLAG–TUBE proteins were eluted by reduced glutathione (10 mM) and FLAG peptide (250 μg/ml), respectively.

**In vivo ubiquitination assay.** Cells were lysed with radioimmunoprecipitation assay buffer containing 50 mM Tris-HCl (pH 7.6), 150 mM NaCl, 1% NP-40, 0.1% sodium dodecyl sulfate (SDS), 0.25% sodium deoxycholate, 10 mM iodoacetamide (Sigma-Aldrich), 10 mM N-ethylmaleimide (Sigma-Aldrich), 0.5 mM 4-(2-aminoethyl)-benzenesulfonyl fluoride hydrochloride (AEBSF), 10 μM MG132, and PhosStop phosphatase inhibitors (Sigma-Aldrich). Endogenous

Cyclin A2, TFIIB, PRPS1/2/3, CUL5, ACSL4, PCNA, DHPS, and BLVRA were immunoprecipitated with an anti-Cyclin A2 antibody (B-8, sc-271682, SCBT), anti-TFIIB antibody (D-3, sc-271736, SCBT), anti-PRPS1/2/3 antibody (A-11, sc-376440, SCBT), anti-CUL5 antibody (sc-13014, SCBT), anti-ACSL4 antibody (22401-1-AP, Proteintech), anti-PCNA antibody (PC10, sc-56, SCBT), anti-DHPS antibody (A-10, sc-365077, SCBT), and anti-BLVRA antibody (F-1, sc-393385, SCBT) respectively and immunoblotted. Alternatively, His$_6$-ubiquitin was introduced into cells and the cells were lysed with a buffer containing 50 mM Tris-HCl (pH 7.6), 300 mM NaCl, 8 M urea, 0.5% Triton X-100, 10 mM iodoacetamide (Sigma-Aldrich), 10 mM N-ethylmaleimide (Sigma-Aldrich), 0.5 mM AEBSF, 10 µM MG132, and PhosStop phosphatase inhibitors (Sigma-Aldrich). His$_6$-ubiquitin-conjugated proteins were pulled down with Probond resin (Thermo Scientific) and immunoblotted with several antibodies.

**Flow cytometry**. Cells were fixed in ice-cold 70% ethanol for at least 30 min and were incubated with a solution containing 0.25 mg/ml RNase A in PBS for 20 min. Then the cells were stained with a solution containing 0.05 mg/ml 7-AAD in PBS and were analyzed using a FACSCanto II flow cytometer (BD Biosciences). Data were obtained by using FACSDiva software (BD Biosciences).

**Protein degradation assay with cycloheximide**. Cells were cultured with cyclo-heximide (Sigma) at a concentration of 50 µg/ml and then incubated for various periods. Cell lysates were then subjected to SDS polyacrylamide gel electrophoresis and immunoblot analysis with anti-cyclin A2, anti-TFIIB, anti-TRIM28, and anti-GAPDH antibodies.

**Statistics and reproducibility**. Data points represent biological replicates. Comparisons between groups were determined using Student's $t$ test or Fisher's exact test. Data were considered statistically significant at $p < 0.05$.

## Data availability

The mass spectrometric datasets were available in ProteomeXchange under the accession number PXD020658 via the jPOST repository. The uncropped images of western blots are shown in Supplementary Fig. 6. All relevant datasets generated during current study are available from the corresponding authors on reasonable request.

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

## Acknowledgements

We thank St. Jude Children's Research Hospital for providing the lentiviral vector system and Siho Sakata, Takahiro Asanuma and Yumi Matsuzaki for help in preparing the paper. This work was supported in part by KAKENHI (17K19506, 18H02607, and 19H05280 to S.H.; 15K15058, 16H06221, 17H05989, and 19K22408 to M.W.) from the Ministry of Education, Culture, Sports, Science and Technology in Japan and by Astellas Foundation for Research on Metabolic Disorders (M.W.), GSK Japan Research Grant 2015 (M.W.), Nakatani Foundation for advancement of measuring technologies in biomedical engineering (M.W.), The Akiyama Life Science Foundation (M.W.), Takeda

Science Foundation (M.W. and S.H.), Futaba Electronics Memorial Foundation (M.W.), the Nagao Memorial Fund (M.W.), the Nakatomi Foundation (M.W.), SENSHIN Medical Research Foundation (M.W.), the NOASTEC Foundation (M.W.), Japan Foundation for Applied Enzymology (S.H.), Tokyo Biochemical Research Foundation (S.H.), Project Mirai Research Grants (S.H.), and Astellas Research Support (S.H.).

## Author contributions

M.W.: Conception and design, acquisition of data, analysis and interpretation of data, drafting or revising the article; Y.S.: Conception and design, analysis and interpretation of data, drafting or revising the article; H.T., F.O., T.K., and Y.Y.: Conception and design, drafting or revising the article; Y.K.: Acquisition of data, analysis and interpretation of data; H.Y., M.S., and H.I.: Drafting or revising the article; K.T.: Analysis and interpretation of data, drafting or revising the article; S.H.: Conception and design, analysis and interpretation of data, drafting or revising the article.

## Competing interests

The authors declare no competing interests.
