## [Peer Review File · Communications Biology]

Reviewers' comments:

Reviewer #1 (Remarks to the Author):

This manuscript authored by Watanabe, Hatakeyama, et al. describes a new modification of the E3 ligase trapping method to identify ubiquitination substrates. The authors engineered two E3 enzymes, Parkin and TRIM28, fused with TUBE, instead of the original substrate-trapping fusion with a single UBA domain, to enhance substrate trapping efficiency and protect substrates from deubiquitination and degradation. The efficiency of identifying substrates appears to be higher than that of the original method. Extensive data are presented from comparative analyses of the new method as opposed to separate expression of TUBE and E3. They also studied effects of differential positions of TUBE fusion on Parkin and TRIM28. Subsequently, a couple of newly identified substrates of Parkin and TRIM28 are validated and the biological significance of the E3-substrate relationships is discussed. The idea to use TUBE instead of a UBA domain is straightforward and the experiments are generally well controlled. Since identifying direct substrates of each E3 remains challenging, introducing a new approach to the field is clearly of high significance. However, the follow-up studies to characterize new substrates are somewhat cursory, lacking solid evidence for the pathways the authors propose.

Specific comments:

Fig. 2 b and c: There are substantial numbers of substrates that were identified only by mono-UBA-fused E3s, i.e., 5 and 3 substrates for FLAG-UBA-Parkin and FLAG-UBA-TRIM28, respectively. Does this suggest structural impact of TUBE on substrate trapping that is not the case for mono-UBA? How physiological are those substrates identified specifically by mono-UBA-E3s?

Fig. S2a and b: More Parkin substrates were identified in HeLaS3 cells than in HEK293T, while the opposite is the case for the TUBE-TRIM28 trapping. What about expression levels of FLAGTUBE-Parkin and FLAGTUBE-TRIM28 in those two cells? Direct immunoblotting for FLAG would allow direct comparison.

Fig. 3d: Effects of shTRIM28 on TUBE-trapped ubiquitinated forms of TFIIB, Cyclin A2 and ATP6V1C1 do not appear compelling, despite the high knockdown efficiency. It would be more convincing if more straightforward methods had been used to detect polyubiquitinated forms, such as immunoprecipitation of each substrate followed by immunoblotting for Ub. The data on cell cycle-associated changes in Cyclin A2 regulation are not compelling, either. A critical question is whether TRIM28-mediated ubiquitination of Cyclin A2 is independent of the major APC/C-dependent mechanisms. To address the question, the authors should determine whether TRIM28-specific ubiquitination site(s) of Cyclin A2 are different from those for APC/C.

Aphidicolin treatment increases steady-state levels of TRIM28 (Fig. 5C). It suggests that TRIM28 expression itself is cell cycle-regulated, which should be commented and discussed in the context of the literature and the authors' experimental data.

The data in Fig. S4 demonstrating TRIM28-mediated repression of FZR1 is interesting but the mechanism shown in panel c is largely speculative. There is no evidence indicating that altered FZR1 levels in TRIM28-depleted cells play a role in Cyclin A2 degradation. The changes in FZR1 protein and mRNA might simply reflect altered cell cycle distribution. The authors' hypothesis does require data from cells with dual depletion of TRIM28 and FZR1.

Given the extreme high affinity of TUBE with polyubiquitinated proteins, this modified method might identify not only direct substrates of an E3 but also proteins in or near its interactome that are ubiquitinated independently of the E3.

Reviewer #2 (Remarks to the Author):

The manuscript submitted by Watanabe et al describes a novel approach for identifying E3 ligase substrates by combining ligase trapping with TUBE-based methods to identify new substrates for Parkin and TRIM28, and the effect of binding and ubiquitination of novel substrates upon mitochondrial depolarization (parkin) and cell cycle progression (TRIM28). Using this technique, the authors were able to determine a new mechanism for TRIM28 in its ability to regulate cyclin A levels at G1/S phase and S phase re-entry, and stabilizing TFIIB and repressing APC(FZR1). The study is extremely well designed and the paper itself is written very well which made reviewing it a pleasure – each time I thought of an experiment or control the authors were able to address this as I was reading and reviewing throughout. The only comments are minor in nature and does not affect my overall decision.

- Because the FLAGTUBE + HA-Parkin/TRIM28 and FLAGTUBE-Parkin/TRIM28 identified different putative substrates as seen in Fig 1f (particularly for TRIM28) can the authors suggest why that is the case? The authors have alluded to having the TUBE in the N-terminus and C-terminus identifying different substrates due to the catalytic domains of parkin at the C-terminus and thereby identifying different substrates. The peptide data is available, have the authors attempted to see whether there is a consensus motif that has preferential binding and ubiquitination? E.g. do the Ub sites on SET, MRT04, SMARCA2, ELAC2 share a similar ubiquitination motif? Or bind to a different region of TRIM28 and thereby not being identified by the fused FLAG-TUBE-TRIM28 which suggests spatial distance plays a role in substrate selectivity.

- There were more substrates candidates of TRIM28 identified in HEK293 compared to Hela while there were more substrates of parkin in Hela compared to HEKs – can the authors elaborate this intriguing result? It would be expected that there would be more substrates in Hela cells for TRIM28 considering the cell cycle proteins in these cancerous cells are constitutively being made and degraded through different stages of the cell cycle.

- Spelling mistake line 336 and 341

- I had a look through the Tables and I would've expected Ser65 on parkin to be phosphorylated by PINK1, especially during CCCP treatment for depolarization. Do the authors have a phospho-blot of Ser65 on parkin to show that it is indeed activated upon CCCP treatment?

- With regards to the mass spectrometry searches, CID was used on an ELITE instrument which allows for a fragment tolerance of 0.6 Da which is appropriate, more recently for Orbitrap instruments HCD is more often used with a lower fragment tolerance for searches - is there a scientific reason for the authors choosing CID? Have the authors tried searching with PD2.4? There have been slight modifications to the Sequest search algorithm which may improve Ub site identification.

- MS data needs to be deposited in a repository such as ProteomeXchange for reviewer/reader access.

Reviewer #3 (Remarks to the Author):

In this paper, the authors describe a method to fuse a TUBE and FLAG tag to two different E3 ligases (Parkin and Trim28) which allows to pull down potential substrates from cell lines. The general idea behind this is very sound and it looks very encouraging. There are a few points though that the authors should consider:

1. Modifications of Parkin have been shown to affect its activity (Chaugule et al 2011, EMBO J, and Burchell et al., PLoS ONE 2012). Considering that the model used here (CCCP treatment, overexpression) is highly artificial anyway, the authors should at least discuss this.
2. The mass spectrometry is disappointingly using PSMs as a quantitative readout. There are much better ways such as peptide and protein intensities.
3. The paper may benefit from someone looking over the language. It sometimes reads a little difficult.

Response to the reviewers:

The reviewers' comments are listed below in bold type. Our responses to each of their comments are shown in regular type.

Reviewer #1 (Remarks to the Author):

This manuscript authored by Watanabe, Hatakeyama, et al. describes a new modification of the E3 ligase trapping method to identify ubiquitination substrates. The authors engineered two E3 enzymes, Parkin and TRIM28, fused with TUBE, instead of the original substrate-trapping fusion with a single UBA domain, to enhance substrate trapping efficiency and protect substrates from deubiquitination and degradation. The efficiency of identifying substrates appears to be higher than that of the original method. Extensive data are presented from comparative analyses of the new method as opposed to separate expression of TUBE and E3. They also studied effects of differential positions of TUBE fusion on Parkin and TRIM28. Subsequently, a couple of newly identified substrates of Parkin and TRIM28 are validated and the biological significance of the E3-substrate relationships is discussed. The idea to use TUBE instead of a UBA domain is straightforward and the experiments are generally well controlled. Since identifying direct substrates of each E3 remains challenging, introducing a new approach to the field is clearly of high significance. However, the follow-up studies to characterize new substrates are somewhat cursory, lacking solid evidence for the

pathways the authors propose.

Specific comments:

1. Fig. 2 b and c: There are substantial numbers of substrates that were identified only by mono-UBA-fused E3s, i.e., 5 and 3 substrates for FLAG-UBA-Parkin and FLAG-UBA-TRIM28, respectively. Does this suggest structural impact of TUBE on substrate trapping that is not the case for mono-UBA? How physiological are those substrates identified specifically by mono-UBA-E3s?

In April of this year (2020), PD2.4, which is much better in LFQ than PD1.4, was introduced into our mass spectrometry pipeline in our laboratory. Since it was suggested by other reviewers to use PD2.4, we re-analyzed all data with PD2.4 and used protein/peptide abundance as a quantitative readout (Page 8, lines 134-136; Page 27, lines 506-509). We show the results as new figures (Fig. 1f, g, Fig. 2d, e, Fig. 3d, e, Supplementary Fig. 1c, d, Supplementary Fig. 2c, and e). As the results of re-analysis, 4 candidates including GORASP2, HSPB1, CASC5, and ACSL4 were detected only in cells expressing mono-UBA-fused Parkin and 3 candidates including CUL5, DDI2, and PRPS2 were detected in cells expressing mono-UBA-fused TRIM28. We further examined the ubiquitination levels of GORASP2, HSPB1, ACSL4, CUL5, DDI2, and PRPS2. We could find antibodies against ACSL4, CUL5, and PRPS2 available for immunoprecipitation and showed that Parkin overexpression or TRIM28 knockdown did not affect the ubiquitination levels of these candidates (new Supplementary Fig. 4a, b and e). On the other hand, we used mouse anti-GORASP2 antibody (sc-271840, SCBT), mouse anti-HSPB1 antibody (sc-13132, SCBT), and mouse anti-DDI2 antibody

(sc-514004, SCBT) for immunoprecipitation of GORASP2, HSPB1 and DDI2, but these antibodies could not immunoprecipitate each candidate protein. For DDI2, although it is a more artificial method, we introduced His₆-ubiquitin into cells, pulled down with Ni-NTA, and performed immunoblotting with the substrate antibody and found that TRIM28 knockdown reduced the amount of ubiquitinated DDI2 (new Supplementary Fig. 4f) (Page 14, lines 256-259). Although there are still candidates that have not been validated, we showed that some proteins identified only by using FLAG-monoUBA-fused E3 are physiological substrates in this revised manuscript.

2. Fig. S2a and b: More Parkin substrates were identified in HeLaS3 cells than in HEK293T, while the opposite is the case for the TUBE-TRIM28 trapping. What about expression levels of FLAGTUBE-Parkin and FLAGTUBE-TRIM28 in those two cells? Direct immunoblotting for FLAG would allow direct comparison.

We thank the reviewer for raising this issue. We examined the expression levels of FLAGTUBE-Parkin and FLAGTUBE-TRIM28 in HEK293T and HeLaS3 cells by immunoblotting. We found that the expression levels of these proteins were comparable in these two cell lines (new Supplementary Fig. 2f, g: input).

3. Fig. 3d: Effects of shTRIM28 on TUBE-trapped ubiquitinated forms of TFIIB, Cyclin A2 and ATP6V1C1 do not appear compelling, despite the high knockdown efficiency. It would be more convincing if more straightforward methods had been used to detect polyubiquitinated forms, such as immunoprecipitation of each substrate followed by immunoblotting for Ub. The data on cell cycle-associated

changes in Cyclin A2 regulation are not compelling, either. A critical question is whether TRIM28-mediated ubiquitination of Cyclin A2 is independent of the major APC/C-dependent mechanisms. To address the question, the authors should determine whether TRIM28-specific ubiquitination site(s) of Cyclin A2 are different from those for APC/C.

In accordance with the reviewer's suggestion, we lysed cells with RIPA buffer, immunoprecipitated substrates by each antibody and then performed immunoblotting with an anti-ubiquitin antibody. We could find antibodies against CCNA2 and TFIIB that were available for immunoprecipitation and found that TRIM28 knockdown reduced the ubiquitination levels of these candidates (new Supplementary Fig. 3a, b). On the other hand, we used rabbit anti-ATP6V1C1 antibody (16054-1-AP, Proteintech) and mouse anti-ATP6V1C1 antibody (sc-271077, SCBT) for immunoprecipitation of ATP6V1C1, but both antibodies could not immunoprecipitate ATP6V1C1. Therefore, although it is a more artificial method, we introduced His₆-ubiquitin into cells, pulled down with Ni-NTA and performed immunoblotting with the substrate antibody and found that TRIM28 knockdown reduced the amount of ubiquitinated ATP6V1C1, in addition to reduction of the amount of ubiquitinated TFIIB (new Supplementary Fig. 3c).

4. Aphidicolin treatment increases steady-state levels of TRIM28 (Fig. 5C). It suggests that TRIM28 expression itself is cell cycle-regulated, which should be commented and discussed in the context of the literature and the authors' experimental data.

We thank the reviewer for raising this issue. Destabilization of TRIM28 protein by aphidicolin treatment was reproducibly observed in our experiments (Fig. 5c). We speculate that other E3(s) promote ubiquitin-dependent degradation of TRIM28 or that self-ubiquitination activity of TRIM28 is increased by aphidicolin treatment. Although the detailed mechanism remains to be elucidated, we are interested in this phenomenon as a future research theme. We described this finding in the Results section (Page 16, lines 298-301).

5. The data in Fig. S4 demonstrating TRIM28-mediated repression of FZR1 is interesting but the mechanism shown in panel c is largely speculative. There is no evidence indicating that altered FZR1 levels in TRIM28-depleted cells play a role in Cyclin A2 degradation. The changes in FZR1 protein and mRNA might simply reflect altered cell cycle distribution. The authors' hypothesis does require data from cells with dual depletion of TRIM28 and FZR1.

We thank the reviewer for appropriate and constructive suggestions. In response to the reviewers' suggestion that the changes in FZR1 protein and mRNA might simply reflect an altered cell cycle distribution, we examined whether the expression level of FZR1 in each cell cycle was changed by TRIM28 expression (see the figure below). We found that TRIM28 knockdown still increased FZR1 protein levels in an asynchronous condition, but there were no significant changes in other phases of the cell cycle. We also found that FZR1 expression was lowest in the G1 phase and high in other phases of the cell cycle. Therefore, as the reviewer pointed out, we concluded that asynchronous

TRIM28 knockdown cells showed higher FZR1 expression level because TRIM28 knockdown in an asynchronous condition causes an increase in the number of cells that exist in the phases (G1 and G1/S) in which the expression level of FZR1 is higher than that in control cells. Based on these findings, we did not perform an experiment using dual deletion of TRIM28 and FZR1 and we deleted the description and figures for the regulation of FZR1 expression by TRIM28 in the revised manuscript. We greatly appreciate the comments by Reviewer 1 that enable us to avoid our wrong interpretations in the first submission.

Supplemental Figure. FZR1 expression in each cell cycle phase of TRIM28-knockdown HEK293T cells. Immunoblot analysis of FZR1 and TRIM28 in each cell cycle phase of TRIM28-knockdown HEK293T cells (top). The intensity of the FZR1 bands was normalized to that of the corresponding GAPDH bands and is indicated as relative intensity of the normalized value of control shRNA-treated cells (bottom). The data represent means \pm s.d. from three independent experiments. AS, asynchronous.

6. Given the extreme high affinity of TUBE with polyubiquitinated proteins, this modified method might identify not only direct substrates of an E3 but also proteins in or near its interactome that are ubiquitinated independently of the E3.

As the reviewer pointed out, TUBE possesses high affinity to polyubiquitinated proteins.

So our method may capture polyubiquitinated proteins near the probes. Therefore, it is important to remove those proteins using adequate negative controls. We used the following strategies (negative controls) to determine specific ubiquitinated peptides identified for each E3 ligase: (1) a FLAG-TUBE probe lacking an E3 ligase protein, (2) a FLAG-TUBE-fused Parkin probe under unstimulated conditions in which target proteins should not be ubiquitinated (Supplementary Table 1), and (3) a probe fused with an E3 ligase with deletion of enzyme activity (Page 8, lines 129-134).

Reviewer #2 (Remarks to the Author):

The manuscript submitted by Watanabe et al describes a novel approach for identifying E3 ligase substrates by combining ligase trapping with TUBE-based methods to identify new substrates for Parkin and TRIM28, and the effect of binding and ubiquitination of novel substrates upon mitochondrial depolarization (parkin) and cell cycle progression (TRIM28). Using this technique, the authors were able to determine a new mechanism for TRIM28 in its ability to regulate cyclin A levels at G1/S phase and S phase re-entry, and stabilizing TFIIIB and repressing APC(FZR1). The study is extremely well designed and the paper itself is written very well which made reviewing it a pleasure – each time I thought of an experiment or control the authors were able to address this as I was reading and reviewing throughout. The only comments are minor in nature and does not affect my overall decision.

1. Because the FLAGTUBE + HA-Parkin/TRIM28 and FLAGTUBE-Parkin/TRIM28 identified different putative substrates as seen in Fig 1f (particularly for TRIM28) can the authors suggest why that is the case? The authors have alluded to having the TUBE in the N-terminus and C-terminus identifying different substrates due to the catalytic domains of parkin at the C-terminus and thereby identifying different substrates. The peptide data is available, have the authors attempted to see whether there is a consensus motif that has preferential binding and ubiquitination? E.g. do the Ub sites on SET, MRTO4, SMARCA2, ELAC2 share a similar ubiquitination motif? Or bind to a different region of TRIM28 and thereby not being identified by the fused FLAG-TUBE-TRIM28 which suggests spatial distance plays a role in substrate selectivity.

In April of this year (2020), PD2.4, which is much better in LFQ than PD1.4, was introduced into our mass spectrometry pipeline in our laboratory. Therefore, we decided to re-analyze all data with PD2.4 and use protein/peptide abundance as a quantitative readout. We show the results as new figures (Fig. 1f, g, Fig. 2d, e, Fig. 3d, e, Supplementary Fig. 1c, d, Supplementary Fig. 2c, and e) (Page 8, lines 134-136; Page 27, lines 506-509). As the results of re-analysis with PD2.4, TXLNA was detected only in cells expressing FLAGTUBE + HAParkin and 7 candidates including PRPS2, FABP5, PCNA, DEK, PDLIM5, DHPS and BLVRA were detected in cells expressing FLAGTUBE + HATRIM28. We performed a motif analysis using MEME Suite on the latter sequences around the ubiquitination site and found no statistically significant motifs. Furthermore, we performed the same analysis in the whole sequences of the 7

candidates and found no statistically significant motifs either. The same analysis was performed for the results (SET, MRTO4, SMARCA2, and ELAC2) obtained with PD1.4, but a statistically significant motif could not be extracted. We also examined ubiquitination levels of PRPS2, CUL5, PCNA, DHPS and BLVRA and revealed that these protein candidates were false positives as ubiquitinated substrates (new Supplementary Fig. 4a, b, c, d) (Page 14, lines 256-259). Therefore, we speculate that some of the proteins identified by only using FLAGTUBE + E3 are unphysiological candidates.

2. There were more substrates candidates of TRIM28 identified in HEK293 compared to Hela while there were more substrates of parkin in Hela compared to HEKs – can the authors elaborate this intriguing result? It would be expected that there would be more substrates in Hela cells for TRIM28 considering the cell cycle proteins in these cancerous cells are constitutively being made and degraded through different stages of the cell cycle.

We thank the reviewer for raising this issue. We speculate two putative reasons for the results. One is that the expression level and/or type of substrates for each E3 may be different in each cell, and another is that the enzymatic activity of each E3 may be different in each cell. It has been reported that the E3 ligase activity of TRIM28 is greatly increased in the presence of some MAGE proteins and that there is a difference in the type of MAGE proteins expressed in each cell. Since the expression levels of the probe used in this study were similar in HEK293T and HeLaS3 cells (new Supplementary Fig. 2f, g; input), our results may indicate that MAGE proteins

expressed in HEK293T cells induce greater activation of TRIM28 than do those expressed in HeLaS3 cells.

3. Spelling mistake line 336 and 341

We thank the reviewer for pointing out our mistakes (from “idendify” to “identify”; from “menbrane” to “membrane”). We corrected the spelling mistakes in the revised manuscript (Page 18, line 338; Page 19, line 343).

4. I had a look through the Tables and I would've expected Ser65 on parkin to be phosphorylated by PINK1, especially during CCCP treatment for depolarization. Do the authors have a phospho-blot of Ser65 on parkin to show that it is indeed activated upon CCCP treatment?

We thank the reviewer for raising this issue. We found an anti-parkin p-Ser65 antibody (Parkin phosphor-Ser65 antibody, orb312554, Biorbyt) that is now commercially available. We purchased this antibody and used it for immunoblotting, but we could not detect p-Ser65 Parkin that should be induced by CCCP treatment. In the product description of this antibody, there are no data of immunoblotting under Parkin activation. So we could not determine whether this antibody can specifically recognize pSer65 Parkin. In our first submitted manuscript, we immunoprecipitated samples with an anti-FLAG antibody, trypsinized them, and then re-immunoprecipitated them with an anti-diGly antibody. The peptide containing Ser65 of mouse Parkin is ELPNHLTVQNCDLEQQpSIVHIVQR or VIFAGK(GG)ELPNHLTVQNCDLEQQpSIVHIVQR. The former was excluded during

the purification process because it does not contain a lysine residue to which ubiquitin is conjugated and the latter was not identified probably because of no ubiquitination via the Lys residue of this peptide. Therefore, we could not correctly evaluate p-Ser65 from the data including data presented in the first submitted manuscript. For the revised manuscript, we performed immunoprecipitation with an anti-FLAG antibody without using an anti-diGly antibody for re-purification and performed analysis with mass spectrometry (new Supplementary Fig. 2b). Comparing with peptide abundance, the unphosphorylated peptide ELPNHLTVQNCDLEQQSIVHIVQR was detected to similar extents in CCCP-untreated and treated cells, while the phosphorylated peptide ELPNHLTVQNCDLEQQpSIVHIVQR was detected only in CCCP-treated cells. Therefore, we concluded that Parkin was actually phosphorylated on Ser65 by PINK1 upon CCCP treatment (Page 11, lines 203-Page 12, line 205).

5. With regards to the mass spectrometry searches, CID was used on an ELITE instrument which allows for a fragment tolerance of 0.6 Da which is appropriate, more recently for Orbitrap instruments HCD is more often used with a lower fragment tolerance for searches - is there a scientific reason for the authors choosing CID? Have the authors tried searching with PD2.4? There have been slight modifications to the Sequest search algorithm which may improve Ub site identification.

In the ELITE instrument, CID can be used both in ion trap mode and FT mode, but HCD can be used only in FT mode. Due to the difference in scan speeds, more data can be acquired in ion trap mode than in FT mode. We prioritized the amount of data and

used the combination of CID/ion trap mode.

As mentioned above, we performed re-analysis with PD2.4 and used protein/peptide abundance as a quantitative readout. We show the results as new figures (Fig. 1f, g, Fig. 2d, e, Fig. 3d, e, Supplementary Fig. 1c, d, Supplementary Fig. 2c, and e).

6. MS data needs to be deposited in a repository such as ProteomeXchange for reviewer/reader access.

In accordance with the reviewer's suggestion, the mass spectrometric datasets were deposited in ProteomeXchange under the accession number PXD020658 via the jPOST repository (Page 31, lines 603-605).

Reviewer #3 (Remarks to the Author):

In this paper, the authors describe a method to fuse a TUBE and FLAG tag to two different E3 ligases (Parkin and Trim28) which allows to pull down potential substrates from cell lines.

The general idea behind this is very sound and it looks very encouraging. There are a few points though that the authors should consider:

1. Modifications of Parkin have been shown to affect its activity (Chaugule et al 2011, EMBO J, and Burchell et al., PLoS ONE 2012). Considering that the model

used here (CCCP treatment, overexpression) is highly artificial anyway, the authors should at least discuss this.

We are grateful to the reviewer for raising this issue. In the Discussion section, we stated that the method used in this study is artificial and that the results must be carefully interpreted. In this study, we screened substrates with a probe in which TUBE was fused to an E3 ligase at its amino or carboxyl terminus. It has been reported that epitope tagging of some E3 ligases affects the activity and stability of E3s. In Parkin, N-terminal epitope tagging results in the release of autoinhibition by its Ubl domain and increase of its enzymatic activity (Chaugule et al. *EMBO J* **30**, 2853-2867 (2011); Burchell et al., *PLoS ONE* **7**, e34748 (2012)). We agree that the use of CCCP may also induce artificial activation, far from physiological conditions. Therefore, substrate candidates identified by our method should be carefully validated and interpreted, considering various factors such as modifications of bait proteins and artificial stimulations in cells (Page 20, lines 373-380).

2. The mass spectrometry is disappointingly using PSMs as a quantitative readout. There are much better ways such as peptide and protein intensities.

In April of this year (2020), PD2.4, which is much better in LFQ than PD1.4, was introduced into our mass spectrometry pipeline in our laboratory. Therefore, we decided to re-analyze all data with PD2.4 and use protein/peptide abundance as a quantitative readout. We show the results as new figures in the revised manuscript (Fig. 1f, g, Fig. 2d, e, Fig. 3d, e, Supplementary Fig. 1c, d, Supplementary Fig. 2c, and e) (Page 8, lines

134-136; Page 27, lines 506-509).

3. The paper may benefit from someone looking over the language. It sometimes reads a little difficult.

We thank the reviewer for the kind advice. We again checked the manuscript throughout and further received English proofreading services. If there are typographical errors, grammatical mistakes or sentences that are difficult to read, please do not hesitate to let us know.

REVIEWERS' COMMENTS:

Reviewer #1 (Remarks to the Author):

The authors responded properly to all my concerns and revised the manuscript appropriately and extensively. I think the revised manuscript is ready for publication after editorial checks.

Reviewer #2 (Remarks to the Author):

After a second review, the authors have made big improvements to the manuscript particularly with the change of proteomics software to improve their identifications and subsequently additional follow up validation experiments. The authors have addressed all the major concerns that I raised in my initial review. Lastly, I would like to praise the authors for a great job in substantially improving their manuscript which will be of big interest in the ubiquitinomics field.

Reviewer #3 (Remarks to the Author):

I am happy with the revision.